# Anti-Angiogenic Property of Free Human Oligosaccharides

**DOI:** 10.3390/biom11060775

**Published:** 2021-05-21

**Authors:** Boram Bae, Haeun Kim, Hyerin Park, Young Jun Koh, Sung-Jin Bae, Ki-Tae Ha

**Affiliations:** 1Department of Korean Medicine, School of Korean Medicine, Pusan National University, Yangsan 50612, Gyeongsangnam-do, Korea; corona1814@pusan.ac.kr (B.B.); haeunhamin@pusan.ac.kr (H.K.); 2Korean Medical Research Center for Healthy Aging, Pusan National University, Yangsan 50612, Gyeongsangnam-do, Korea; rin8998@pusan.ac.kr; 3GI Innovation, Inc., A-1116, Tera Tower, Songpa-gu, Seoul 05855, Korea; youngjun.koh@gi-innovation.com

**Keywords:** human milk oligosaccharides, sialyllactose, angiogenesis, VEGFR-2, inhibitor

## Abstract

Angiogenesis, a fundamental process in human physiology and pathology, has attracted considerable attention owing to its potential as a therapeutic strategy. Vascular endothelial growth factor (VEGF) and its receptor (VEGFR) are deemed major mediators of angiogenesis. To date, inhibition of the VEGF-A/VEGFR-2 axis has been an effective strategy employed in the development of anticancer drugs. However, some limitations, such as low efficacy and side effects, need to be addressed. Several drug candidates have been discovered, including small molecule compounds, recombinant proteins, and oligosaccharides. In this review, we focus on human oligosaccharides as modulators of angiogenesis. In particular, sialylated human milk oligosaccharides (HMOs) play a significant role in the inhibition of VEGFR-2-mediated angiogenesis. We discuss the structural features concerning the interaction between sialylated HMOs and VEGFR-2 as a molecular mechanism of anti-angiogenesis modulation and its effectiveness in vivo experiments. In the current state, extensive clinical trials are required to develop a novel VEGFR-2 inhibitor from sialylated HMOs.

## 1. Introduction

Milk oligosaccharides are milk components with diverse biological functions [1]. In particular, human milk has been utilized as a medicinal food due to its nutritional composition and non-nutritive bioactive factors [2]. Breast milk, which is ideal for infants, contains numerous complex ingredients, including proteins, lipids, carbohydrates, minerals, and other minor nutrients [3]. Among human milk components in colostrum, oligosaccharides are the third most abundant, present at concentrations of up to 20–25 g/L [4]. However, newborn babies lack enzymes that digest complex milk oligosaccharides. Thus, the precise physiological role of milk oligosaccharides remained elusive until the 1960s [4,5]. However, pioneering studies in the early 20th century have discovered that the carbohydrate fraction of human milk contains growth-promoting factors for Lactobacillus bifidus [4]. Currently, accumulated data suggest that human milk oligosaccharides (HMOs) are prebiotics, as well as modulators of the intestinal mucosal and systemic immune response [4,6].

All HMOs contain five different monosaccharides, including *D*-glucose, *D*-galactose, *L*-fucose, *N*-acetylglucosamine and *N*-acetylneuraminic acid [3]. Approximately 150 different types of oligosaccharides have been identified in human milk, with all possessing a lactose unit at the reducing end [3,5]. Typically, HMOs are classified as neutral and acidic oligosaccharides based on the respective presence or absence of negatively charged *N*-acetylneuraminic acid, that is, sialic acid [6]. In several aspects, the biological activities of acidic HMOs tend to differ from those of neutral HMOs [7,8]. Naturally occurring free oligosaccharides harboring sialic acid have been found in both plasma and urine in healthy men and women, especially in pregnant and lactating women [9,10,11]. Free sialylated oligosaccharides frequently present in human milk inhibit the adhesion of immune cells, cholera toxin, and influenza virus with endothelial or epithelial cells [12,13,14]. Moreover, the sialic acid-containing portion of HMOs is essential for early neurodevelopment and cognition [15,16].

In the present review, we focused on the modulation of angiogenesis via human-derived oligosaccharides, especially free sialylated oligosaccharides, to ameliorate diseases associated with excessive angiogenesis.

## 2. Angiogenesis in Human Health and Disease

### 2.1. Physiological and Pathological Angiogenesis

In vertebrates, the vascular system plays a crucial role in organ homeostasis by transporting oxygen and nutrients [17]. Closed blood vessel systems, like those in vertebrates, first appeared in their common ancestor over 500 million years ago to optimize flow dynamics and barrier function [18]. Reportedly, metabolic requirements for oxygen and nutrients induce new blood vessel formation from the existing ones, a process termed angiogenesis [19]. Angiogenesis was initially considered as a physiological process for maintaining metabolic homeostasis in the field of developmental biology [20]. Inevitably, new blood vessel formation plays a critical role in early development, tissue growth and wound healing [19]. Furthermore, female reproductive physiology, including oocytogenesis, embryo implantation and the menstruation cycle, are regulated by angiogenesis (Figure 1) [21].

Conversely, angiogenesis could be employed as a therapeutic target for treating pathological conditions characterized by either insufficient vascularization or excessive vasculature [22,23]. Tumor-derived factors to promote neovascularization were first postulated in the late 1930s [23]. In 1971, Folkman [24] suggested that inhibition of angiogenesis might have potential therapeutic implications in cancer therapy. These pioneering studies have highlighted the concept that angiogenesis is an important biological process and therapeutic target in diverse diseases, including cancer [25]. The excessive growth of new vessels can aggravate diverse disorders, ranging from cancer and obesity to retinopathy, such as age-related macular degeneration [22,26]. Psoriasis, arthritis, inflammatory bowel disease, benign prostate hyperplasia, endometriosis, ovarian cysts, and uterine bleeding also have been reported to have a mutual correlation with excessive angiogenesis [22,27,28]. In contrast, insufficient angiogenesis can contribute to various diseases, such as stroke, myocardial infarction, diabetic ulcers, atherosclerosis, coronary artery disease, systematic lupus erythematosus, preeclampsia, Alzheimer’s disease, and Crohn’s disease [22,26,29]. Thus, balanced regulation of the angiogenic process might be a key factor for maintaining human health and preventing or treating numerous diseases.

### 2.2. Vascular Endothelial Cell Growth Factors (VEGFs) and Their Receptors (VEGFRs) as Therapeutic Targets for Pathological Angiogenesis

Angiogenesis involves the formation and maintenance of new blood vessels via the cooperation of multiple cells in vascular networks, including vascular endothelial cells, their progenitor cells and pericytes [30,31]. Based on the dynamic interplay between these cells, new blood vessel-like structures are formed via multistep processes, such as sprouting, tip cell migration and tube formation [22,32]. These processes are regulated by signaling between endothelial cells and the perivascular cell layer by secreting growth factors, direct cell–cell interaction, and extracellular matrix production [31,32,33]. Among the secretory growth factors that regulate angiogenesis, VEGFs are the most important, as they play key roles in multiple steps of neovascular formation (Figure 2) [31,34]. Thus, the axis of VEGFs and their receptors (VEGFRs) has been considered a therapeutic target for modulating angiogenesis since the beginning of anti-angiogenic studies [24,35].

In mammals, the VEGF family consists of five members, including VEGF-A, VEGF-B, VEGF-C, VEGF-D, and placental growth factor (PLGF). These ligands bind to their respective receptors (VEGFRs), which belong to the type IV receptor tyrosine kinase (RTK) family and are composed of three members, VEGFR-1, VEGFR-2, and VEGFR-3 [36,37]. Once the ligand binds to the receptor, homo- or hetero-dimeric interactions of VEGFRs initiate the autophosphorylation of intracellular tyrosine residues, as well as downstream signaling pathways responsible for the proliferation, migration and remodeling of the vascular endothelial cells [34,36]. Among them, VEGFR-1 and VEGFR-2 play critical roles in physiological and pathological angiogenesis. In blood vascular endothelial cells, angiogenesis is predominantly mediated via VEGFR-2 activation [38]. VEGF-A, VEGF-B, and PLGF are high-affinity ligands of VEGFR-1, but the kinase activity of VEGFR-1 is relatively weak for the progression of the angiogenesis [39]. In some cancers, VEGF-C sustains VEGFR-2 activation by binding to VEGFR-2 even when inhibiting VEGF-A [40,41]. VEGF-C and VEGF-D stimulate VEGFR-3 activation, which plays an indispensable role in both angiogenesis and lymphangiogenesis [42]. Furthermore, the cooperative signaling between VEGFR-2 and -3 is involved in forming new lymphatic vessels (Figure 3) [43].

Therefore, considerable efforts to inhibit VEGFR-2 activation have been made to suppress angiogenesis as pathogenic angiogenesis is predominantly mediated by the VEGF-A/VEGFR-2 axis [44]. Several strategies for suppressing excessive angiogenesis have been exploited, including neutralizing monoclonal antibodies, VEGF-trapping recombinant proteins and small molecule tyrosine kinase inhibitors [34,35,45]. Among them, neutralizing VEGF-A using the anti-VEGF-A monoclonal antibody, bevacizumab (brand name Avastin), has been inhibiting angiogenesis most successfully in clinical settings. It has been approved as a conventional treatment for several cancers, such as colorectal cancer, lung cancer, glioblastoma, renal cell carcinoma, and age-related macular degeneration [46,47,48]. However, the use of bevacizumab has revealed several limitations, such as high cost, no effect on overall survival in a few cancer cases, and adverse effects on coronary and peripheral artery disease [49,50]. Accordingly, several researchers have focused on developing an effective and safe anti-angiogenic agent from small molecule compounds [34,35]. The majority of small molecule anti-angiogenic agents target the tyrosine kinase activity of VEGFRs [51,52]. However, these chemical inhibitors for tyrosine kinases have been unsuccessful owing to their low specificity and mutation-induced drug resistance [51,53]. As the extracellular ligand-binding region of RTKs is markedly diverse in terms of the protein structure, they are considered more suitable for developing specific inhibitors [54,55]. Accordingly, several investigations reported that interactions between VEGF-A and its receptor could be intercepted by employing recombinant peptides and peptidomimetic chemicals [56,57,58].

## 3. Human Milk Oligosaccharides and Angiogenesis

### 3.1. Oligosaccharides and Angiogenesis

Several different types of oligosaccharide chains exist in mammals as bound to proteins, lipids, or repeating sugar units [59,60]. Oligosaccharide-linked proteins or lipids, that is, glycoproteins or glycolipids, respectively, are mainly located on the cell surface and have distinct biological functions, including mediation of viral/bacterial infection, immune response, cell–cell interaction and cancer progression [61]. Additionally, numerous ligand–receptor interactions might be regulated by the glycosylation status, especially those involving G protein-coupled receptors and growth factor receptors [61,62,63]. It is well known that ligand binding and trafficking of epidermal growth factor receptor (EGFR), fibroblast growth factor-1 and VEGFR-2, the most crucial RTKs in cancer-associated angiogenesis, are affected by site-specific *N*-glycosylation [64,65,66]. Conversely, in the present review, we focused on the pro- or anti-angiogenic roles of unbound oligosaccharides to determine their potential as therapeutic modulators of angiogenesis.

As shown in Table 1, different types of oligosaccharides have pivotal roles in angiogenesis. Hyaluronan types, including their fragments, generally promote angiogenic processes [67,68,69,70,71,72,73,74,75,76,77,78,79]. Several hyaluronan receptors, including CD44, receptor for hyaluronan-mediated motility, LYVE-1 and CD31 [69,70,72,73,76], and their downstream signaling molecules, such as protein kinase C, Src, extracellular signal-regulated kinase (ERK), transforming growth factor-β and Janus kinase/signal transducer and activator of transcription [75,77,78], are involved in hyaluronan-stimulated angiogenesis. Two studies reported that high molecular weight hyaluronan inhibits angiogenesis, whereas small molecular weight hyaluronan promotes angiogenesis [80,81]. Heparin oligosaccharides, such as heparin and their fragments combined with corticosteroid and heparin-like glycosaminoglycans, caused inhibitory effects on angiogenesis [80,82]. However, heparin itself promotes angiogenesis by binding to α2-macroglobulin, thus decreasing the inhibitory effect of α2-macroglobulin on VEGF [83]. Furthermore, fucosylated oligosaccharides typically promote angiogenesis by interacting with FGF-2 and galectin-12 and through the secretion of basic fibroblast growth factor (bFGF) and VEGF [84,85,86]. However, fucosylated glycosaminoglycan and its derivatives suppress angiogenesis by inhibiting heparanase [87].

Among the oligosaccharides listed in Table 1, only some, including lacto-*N*-neotetraose (LNnT), fucosylated glycans, and sialylated oligosaccharides, are classified as HMOs. Especially, LNnT, a linear chain of a tetrasaccharide composed of galactose [88], *N*-acetylglucosamine and lactose, is reportedly a prebiotic that promotes the growth of Bifidobacterium longnum, especially the subspecies infantis [89,90]. Helminths-derived LNnT showed an immunosuppressive effect by augmenting Gr1^+^ cells and inhibiting naïve CD4^+^ cells [91]. Recently, it was reported that LNnT accelerates the wound healing process by inducing angiogenesis and promoting type 2 immune responses [92,93]. Approximately 50–80% of HMOs are fucosylated with fucose linked in α1-2, α1-3 or α1-4 linkages to galactose, glucose, or *N*-acetylglucosamine [2], and have demonstrated beneficial effects on reducing *Campylobacter jejuni*-associated diarrhea in a human translational study [94]. Core fucosylated free oligosaccharides derived from maternal milk *N*-glycosylated proteins activate B cells via B cell receptor-mediated downstream signaling [95]. Several studies have revealed that fucosyltransferases and fucosylated proteins play positive roles in angiogenesis via the activation of fibroblasts, vascular endothelial cells, and endothelial progenitor cells [96,97,98]. In addition, the fucosylated glycans increase angiogenesis by interacting with galectin-12 or releasing angiogenic bFGF and VEGF [84,86].

### 3.2. Roles of Sialylated HMOs in Pathologic Angiogenesis

Several cell-surface proteins, such as mucins, ion channels, receptors, and adhesion molecules, are highly glycosylated with terminal sialic acid residues [61]. Numerous studies have revealed that cell surface glycosylated molecules bound to growth factor receptors regulate their proangiogenic function [99,100,101]. *N*-glycosylation, especially that of terminal sialic acid residues, regulates ligand-dependent activation of VEGFR-2 [102]. Not only proteins, a sialylated glycosphingolipid, GM3, reportedly exhibits an anti-angiogenic effect by inhibiting VEGFR2 activation [103,104]. Although two review papers have discussed the role of glycosylation as a novel therapeutic target for diseases associated with excessive angiogenesis [105,106], none of them described the exact functions of sialylated HMOs in angiogenesis. In 2004, Rudloff et al. [107] demonstrated that sialylated HMOs have an anti-angiogenic effect on bovine vascular endothelial cells by employing an in vitro tube formation assay. However, the study failed to identify the precise components of acidic HMOs and the mechanism underlying their anti-angiogenic effects.

In this regard, our group recently revealed the specific components of anti-angiogenic acidic HMOs and their underlying mechanisms. Unlike other HMOs summarized in Table 1, we identified that 3′- and 6′-sialyllactose inhibited angiogenesis (Figure 4) [108,109]. However, their analogs, 3′-sialyl-*N*-lactosamine and 6′-sialyl-*N*-lactosamine, were unable to bind to VEGFR-2 or suppress their activation, despite the only difference being a single glycan unit, glucose, and *N*-acetylglucosamine [108]. Moreover, our data unraveled that 6′-sialylgalactose is a minimal component that harbors superior binding affinity to VEGFR-2 and suppresses its activation [110]. We also found a potent mechanism that sialyllactose and sialylgalactose might interfere with the interaction between VEGF-A and the immunoglobulin-like domain 2 of VEGFR-2. Binding affinity of free oligosaccharides to VEGFR-2 measured by surface plasmin resonance was relatively lower than that of VEGF-A to VEGFR-2 [110,111]. Among the free sialylated oligosaccharides, 6′-sialylgalactose possesses a higher binding affinity than other oligosaccharides [110].

Following VEGFR-2 inhibition by sialylated oligosaccharides, the downstream signaling molecules, including ERK, Akt and p-38, were also suppressed (Figure 5) [108,110]. Furthermore, administration of sialylated oligosaccharides sufficiently inhibited angiogenesis in allograft cancer, benign prostate hyperplasia, and premature retinopathy models [108,109,110].

As described above, antibody- and recombinant protein-based drugs are superior to small molecules, such as oligosaccharides, in terms of target specificity. However, their clinical application might be limited due to the high cost, the risk of immunogenicity following long-term treatment, and limited accessibility to target pathological foci owing to their large size [110,112]. Natural sialylated HMOs reportedly possess several valuable properties, including low molecular weight, low immunogenicity, and high accessibility to therapeutic targets. Moreover, the safety of 3′- and 6′-sialyllactoses has previously been confirmed for use in infants as well as in the general population, based on rodent and porcine models [113,114,115]. Therefore, these in vivo efficacy and safety assessment studies potentiate sialylated oligosaccharides as an anti-angiogenic agent via suppressing the VEGF-A/VEGFR-2 axis. However, to develop sialylated HMOs as clinically available VEGFR-2 inhibitors, further extensive preclinical studies using animal models of pathological angiogenesis, as well as clinical trials, should be warranted. Moreover, identifying superior oligosaccharide analogs with higher binding affinities might help guarantee improved anti-angiogenic effects.

## 4. Conclusions

In the present review, we highlighted the emerging role of sialic acid-containing HMOs in the suppression of VEGFR-2-mediated angiogenesis. Disparately from other glycans contained in HMOs, sialyllactose, and sialylgalactose could inhibit the activation of VEGFR-2 by binding to its immunoglobulin-like domain 2. Although the anti-angiogenic effects of sialylated HMOs have been evaluated in limited in vivo models, such as several cancers, premature retinopathy, and benign prostate hyperplasia, their anti-angiogenic efficacy still has the potential to cure other pathological conditions associated with excessive angiogenesis. Moreover, extensive clinical trials using sialylated oligosaccharides would lead us to new and novel strategies to develop clinically available VEGFR-2 inhibitors from sialylated HMOs.

## Figures and Tables

**Figure 1 biomolecules-11-00775-f001:**
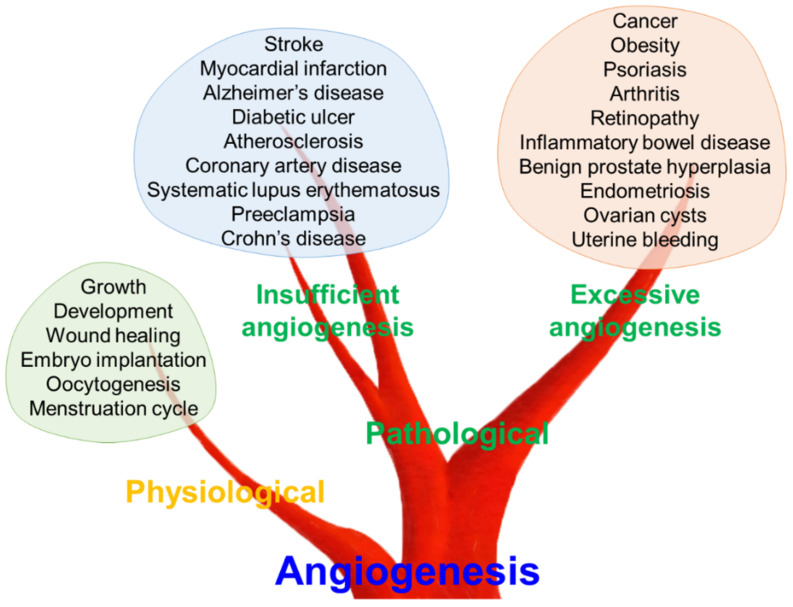
Role of angiogenesis in physiological and pathological conditions. Angiogenesis is a fundamental process in human physiology and pathology. Dysregulated angiogenesis, both insufficient and excessive angiogenesis, can lead to various pathological conditions.

**Figure 2 biomolecules-11-00775-f002:**
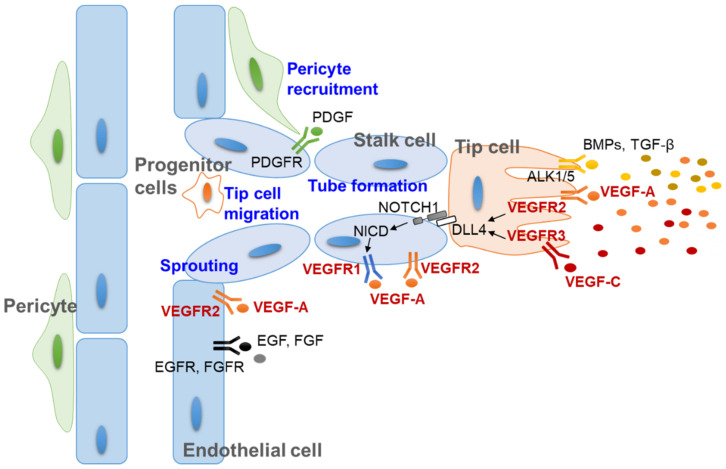
Molecular mechanism of angiogenesis and multicellular interaction during new vessel development. In response to stimulators such as VEGFs, vascular endothelial cells sprout from the basement membrane and migrate to the site of new vessel formation. The formation of tip and stalk cells is regulated by the VEGF-DLL4/NOTCH signaling pathway. The other stimulating molecules, such as PDGF, EGF, FGF, BMPs, and TGF-β, also cooperate to regulate pericytes and tip cells. ALK1/5, actin receptor-like kinase 1/5; BMP, bone morphogenic protein (yellow dots); DLL4, delta-like protein 4; EGF, epidermal growth factor (black dot); EGFR, epidermal growth factor receptor; FGF, fibroblast growth factor (gray dot); FGFR, fibroblast growth factor receptor; NICD, intracellular domain of the notch protein; PDGF, platelet-derived growth factor (green dot); PDGFR, platelet-derived growth factor receptor; TGF-β, transforming growth factor-β (dark yellow dots); VEGF-A, vascular endothelial cell growth factor A (orange dots); VEGF-C, vascular endothelial cell growth factor C (red dots); VEGFR, vascular endothelial cell growth factor receptor.

**Figure 3 biomolecules-11-00775-f003:**
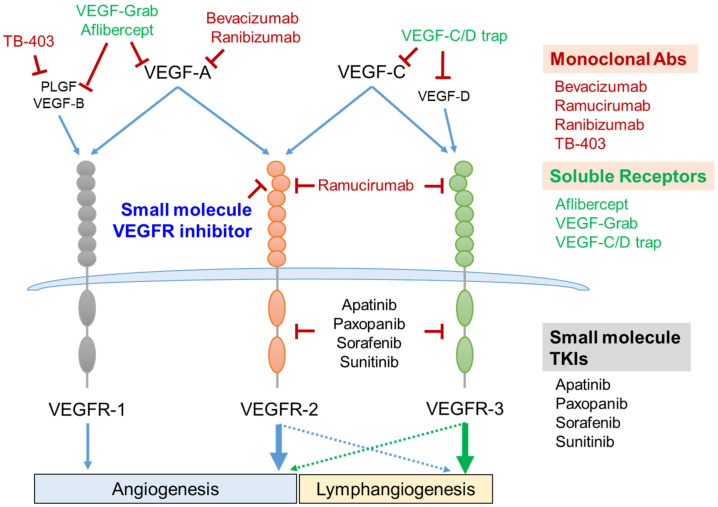
VEGF/VEGFR axis as a therapeutic target and its intervention strategies. Five members of the VEGF family are composed of VEGF-A, VEGF-B, VEGF-C, VEGF-D, and PLGF. VEGF-A binds to both VEGFR-1 and VEGFR-2. VEGF-C binds to both VEGFR-2 and VEGFR-3. VEGF-B and PLGF bind to VEGFR-1, and VEGF-D binds to VEGFR-3. VEGFR-1- and VEGFR-2-mediated signaling cascades regulate vascular angiogenesis. VEGFR-3 activation is essential for lymphangiogenesis (green arrow). VEGFR-2 cooperatively activates lymphangiogenesis with VEGFR-3 (blue dashed arrow), and VEGFR-3 also slightly enhances the vessel angiogenesis (green dashed arrow). Several monoclonal antibodies and recombinant soluble receptors consisting of the extracellular domains of VEGFRs have been successfully developed as therapeutic anti-angiogenic agents. Small molecules targeting the intracellular tyrosine kinase domain or extracellular VEGF-binding domain are under development as novel strategies for inhibiting angiogenesis. Abs, antibodies; PLGF, placenta growth factor; TKIs, tyrosine kinase inhibitors; VEGF, vascular endothelial cell growth factor; VEGFR, vascular endothelial cell growth factor receptor.

**Figure 4 biomolecules-11-00775-f004:**
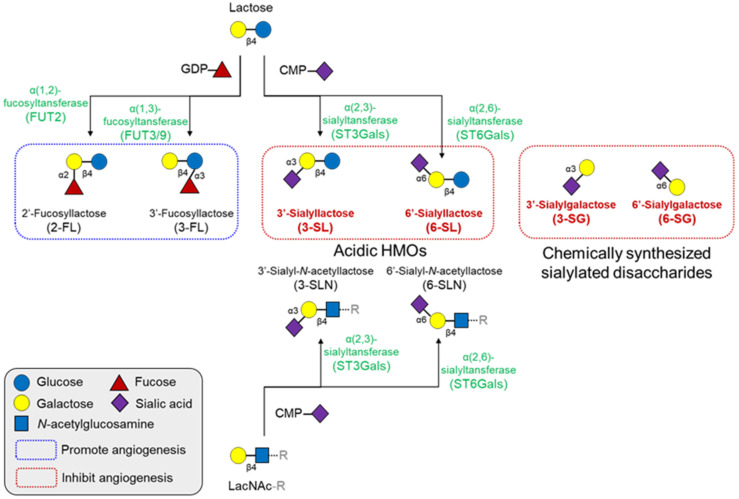
Structures and synthetic pathways of major HMOs and their effects on angiogenesis. All HMOs consist of a lactose core or LacNAc core, with a few exceptions. These cores can be enzymatically elongated in repeats of LacNAc. The elongated HMO chains can be further decorated with fucosylation or sialylation by fucosyltransferases or sialyltransferases, respectively. Fucosylated HMOs generally promotes angiogenesis, but several sialyllactose analogs inhibit angiogenesis. CMP, cytidine monophosphate; GDP, guanosine diphosphate; HMO, human milk oligosaccharides; LacNAc, *N*-acetyllactosamine.

**Figure 5 biomolecules-11-00775-f005:**
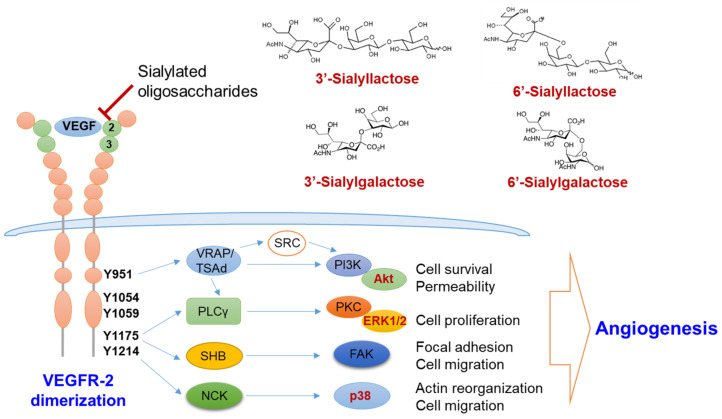
Molecular mechanism of the anti-angiogenic action by sialic acid-containing oligosaccharides. Sialylated lactose and sialylgalactose with sialic acid linked α2-3 or α2-6 to galactose inhibit the activation of VEGFR-2 by interfering the binding between VEGF and VEGFR-2 via Ig-like domain 2 and 3 (green circle). Inhibition of VEGFR-2 activation thereby suppresses the downstream angiogenic signaling pathways, such as the PI3K/Akt, PKC/ERK1/2 and p38 pathways. ERK1/2, extracellular signaling-regulated kinases 1/2; FAK, focal adhesion kinase; NCK, non-catalytic region of tyrosine kinase; PI3K, phosphoinositide 3-kinase; PLCγ, phospholipase γ; PKC, protein kinase C; SHB, SH2 domain-containing adapter protein B; TSAd, T-cell specific adaptor protein; VEGF, vascular endothelial cell growth factor; VEGFR, vascular endothelial cell growth factor receptor; VRAP, VEGF-receptor activated protein.

**Table 1 biomolecules-11-00775-t001:** Summary of studies investigating the effects of oligosaccharides on angiogenesis.

Compound		Reference (PMID)	Exam	Molecular Target	Effect on Angiogenesis	Disease Model
**Heparin, heparan sulfate, or their fragments**		7681826	in vitro	α2-Macroglobulin	Promotion	N.D. ^(1)^
		3746342	in vitro, in vivo	Growth of cerebral microvessel endothelial cell	Inhibition	N.D. ^(1)^
		14517393	in vitro	FGF	Inhibition	N.D. ^(1)^
**Hyaluronan, hyaluronic acid, or their fragments**		2408340	in vitro	N.D. ^(1)^	Promote	N.D. ^(1)^
		2472284	in vitro	Endothelial cell proliferation	Promotion or inhibition ^(2)^	N.D. ^(1)^
		1384133	in vitro	N.D. ^(1)^	Promotion or inhibition ^(2)^	N.D. ^(1)^
		8647630	in vitro	CD44	Promotion	N.D. ^(1)^
		7543630	in vivo	N.D. ^(1)^	Promotion	Skin wound healing
		18544273	in vitro	RHAMM (receptor for hyaluronan mediated motility)	Promotion	Skin wound healing
		12194965	in vitro	PKCα, -β1, -β2, -ε	Promotion	Skin wound healing
		19724912	in vitro	CD44 and RHAMM	Promotion	Wound healing
		19913615	in vivo	LYVE-1 (lymphatic vessel endothelial hyaluronan receptor 1) and CD31	Promotion	Skin wound healing
		16544303	in vitro	Endothelial cell proliferation	Promotion	Wound healing
		27588388	in vivo	Phosphorylation of Src and ERK TGF-β expression	Promotion	Diabetic wound
		31037151	in vivo	macrophage M2 polarization (MAPK, JAK/STAT pathway)	Promotion	Myocardial infarction
		26917404	in vitro, in vivo	CD44	Promotion	N.D. ^(1)^
		19720068	in vitro	VEGF (mRNA level)	Promotion	N.D. ^(1)^
**Lacto-N-Neotetraose ^(3)^**		31969618	in vitro, in vivo	Th2 immune response	Promotion	Skin wound healing
**Sialylated oligosaccharides**	6′-sialylgalactose, 3′-sialylgalactose	31604908	in vitro, in vivo	VEGF receptor 2	Inhibition	Cancer and retinopathy
	6′-sialyllactose, 3′-sialyllactose ^(3)^	28938544	in vitro, in vivo	VEGF receptor 2	Inhibition	Cancer
	6′-sialyllactose ^(3)^	31383249	in vitro, in vivo	VEGF receptor 2	Inhibition	Benign prostatic hyperplasia
**Fucosylated oligosaccharides**	Fucosylated glycosaminoglycan	33667689	in vitro, in vivo	Heparanase	Inhibition	Cancer
	Fucosylated chondroitin sulfate	12496356	in vitro, in vivo	FGF-2	Promotion	Ischemia and thrombosis
	3′-fucosylated glycans ^(3)^	31914594	in vitro, in vivo	Galectin-12	Promotion	Adipose metabolic disorder
	2′-fucosyl lactose (H-2g) ^(3)^	15498849	in vitro, in vivo	Secretion of bFGF and VEGF	Promotion	N.D. ^(1)^

^(1)^ N.D.: not determined. ^(2)^ Promoted by low molecular weight hyaluronan and inhibited by high molecular weight hyaluronan. ^(3)^ Ingredients of human milk. ERK, extracellular signal-regulated kinase; FGF-2, fibroblast growth factor; TGF-β, Transforming growth factor-beta; VEGF, vascular endothelial growth factor.

## Data Availability

The data presented in this study are openly available.

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
