# Peer review of "Anti-Angiogenic Property of Free Human Oligosaccharides"

_biomolecules, 2021, doi:10.3390/biom11060775_

Round 1
Reviewer 1 Report
This review summarizes the processes and mechanisms of angiogenesis, and the effects of oligosaccharides on angiogenesis, particularly, the role of sialic acid-containing HMOs in the suppression of VEGFR-2-mediated angiogenesis.
Below are some comments.
- Figure 1: Excessive angiogenesis can aggravate cancer, obesity, and retinopathy, which was pointed out in the text. However, in Figure 1, excessive angiogenesis is shown to impact psoriasis, arthritis, inflammatory bowel disease, benign prostate hyperplasia, endometriosis, ovarian cysts, and uterine bleeding, which are not described in the text and should be discussed. Also, the corresponding references are lacking.
Samilarly, the role of angiogenesis in atherosclerosis, coronary artery disease, systemic lupus erythematosus, preeclampsia, and Crohn's disease should be discussed and references are needed.
- Figure 2: The dark yellow dots on the rightest in Figure 2 need to be explained in the legend. Anotations of each element involved in the process of new vessel growth are needed.
- Figure 3: Aflibercept and TB-403 can inhibit VEGF-B and PIGF (PMID: 26775688). There are two ‘Ab’ between VEGFR-2 and VEGFR-3. What do they mean?
In the legend, "VEGFR-1- and VEGFR-2-mediated signaling cascades regulate vascular angiogenesis by activating vascular endothelial cells”. However, in the figure, VEGFR-1, VEGFR-2, and VEGFR-3 all have an arrow pointing to angiogenesis. Moreover, ‘VEGFR-3 activation is essential for lymphangiogenesis’. However, in the figure, VEGFR-2 also has an arrow to lymphangiogenesis. These need to be better explained.
- Figure 4: The purple rectangle needs annotation. The word ‘3’-Sialylgalactose’ on the right should be ‘6’-Sialylgalactose’?
- Figure 5: What’s the meaning of number 2 and 3 in the green circle near VEGF?
- Line 85-87, references are needed for ‘In contrast, insufficient angiogenesis can contribute to various diseases, such as stroke, myocardial infarction, diabetic ulcers, and neurodegenerative diseases’.
Author Response
The response was added as a MS word file.

Reviewer 2 Report
Manuscript: Anti-angiogenic property of sialylated human milk oligosaccharides.
This is a well-written comprehensive review over the impact of free oligosaccharides on angiogenesis, which is a rather new promising scientific field. The title is somewhat misleading as only a small part of the review deals specifically with sialylated milk oligosaccharides. A suggestion is therefore to change the title (and the chapter 3 title) from “sialylated human milk oligosaccharides” to “free human oligosaccharides ” The review should also benefit from including information on the naturally occurrence of free sialylated oligosaccharides as they have been found in both plasma and urine in healthy men and women as well as in pregnant and lactating women. See for example these references: Shimada I et al., J Gastroenterol., 1995, 30(1):21-7, Ekman B et al., Glycokonj. J.,2015, 32(8):635-41 and Fu D, Analyt. Biochem., 1999, 269:113-23. It would also be interesting to include more information on affinity of free oligosaccharides towards VEGFRs including a discussion of the relevance and concentrations needed in future treatment options.
Please also correct or respond to following remarks.
Chapter 1 Introduction
Line 39-40: The sentence ending with “…, unlike other disaccharides.” Should be rephrased. The milk oligosaccharide lactose (often not included in the term HMO) is a disaccharide present in human milk and digested in the intenstinal mucosa. Suggestion: “However, newborn babies lack enzymes that digest complex milk oligosaccharides. “
Line 47: The enantiomeric description should be in capital letters (D- or L-).
Line 49: The term “basic receptor molecule” is misleading. Use instead “lactose unit at the reducing end”.
Line 51: Delete “a” in “…absence of a negatively..”. HMOs can have several sialic acid residues.
Line 52: Omitt “terms of”.
Chapter 2
Line 115: Omitt “growth factors”.
Line 124-126: It is unclear if VEGF-A has stimulatory or inhibitory actions on VEGFR-2 and VEGFR-1. Please describe VEGF-A, B and Cs actions more clearly.
Line 126: Delete “,” preceding “and”. This remark includes several additional places in the text.
Chapter 3:
Line 167: Change “as bound forms of proteins” to “as bound to proteins…”
Line 189: Suggests change of “revealed” to “caused”.
Line 189-190: The sentence is unclear. Change to “However, heparin itself promotes angiogenesis by binding to α2-macroglobulin, thus decreasing the inhibitory effect of α2-macroglobulin on VEGF.”
Line 202-204: This sentence is unclear. Suggest changing to “Approximately 50-80 % of HMOs are fucosylated with fucose linked in α1-2, α1-3 or α1-4 linkages to galactose, glucose or N-acetylglucosamine.
Line 206: What is meant by milk N-glycan? N-glycans are mainly proteinbound via Asn and not free HMOs. Suggest omitting this sentence as it is not clearly relevant in this context.
Line 210: What is meant by “soluble form”? All glycans are soluble in water based solutions. Please explain or change.
Table 1
“Lacto-n-Neotetraose” should be “Lacto-N-neotetraose”
“6´-sialygalactose” should be “6´sialylgalactose”
Fig. 4 and line 236: Conflicting descriptions. In the figure the blue square symbol is denoted “N-acetylgalactosamine”. The text describes lactosamine structures containing “N-acetylglucosamine”. N-acetyllactosamine (LacNAc) is a disaccharide consisting of Gal β1-4 linked to GlcNAc (=N-acetylglucosamine). Galβ1-4GalNAc is not a part of free HMOs but exists O-linked to Ser/Thr on proteins. Please correct and explain.
Fig. 5
The first sentence in the figure text should be changed to: “Sialylated lactose and sialylgalactose with sialic acid linked α2-3 or α2-6 to galactose inhibit the action….”
Author Response
The response letter was added as an MS word file.
